# Bioinspired Propulsion System for a Thunniform Robotic Fish

**DOI:** 10.3390/biomimetics7040215

**Published:** 2022-11-28

**Authors:** Iliya Mitin, Roman Korotaev, Artem Ermolaev, Vasily Mironov, Sergey A. Lobov, Victor B. Kazantsev

**Affiliations:** 1Center for Neurotechnology and Machine Learning, Immanuel Kant Baltic Federal University, Kaliningrad 236041, Russia; 2Neurotechnology Department, National Research Lobachevsky State University of Nizhny Novgorod, Nizhny Novgorod 603022, Russia; 3Russian State Scientific Center for Robotics and Technical Cybernetics, St. Petersburg 194064, Russia

**Keywords:** robotic fish, biomorphic system, thunniform locomotion, fish swimming

## Abstract

The paper describes a bioinspired propulsion system for a robotic fish model. The system is based on a combination of an elastic chord with a tail fin fixed on it. The tail fin is connected to a servomotor by two symmetric movable thrusts simulating muscle contractions. The propulsion system provides the oscillatory tail movement with controllable amplitude and frequency. Tail oscillations translate into the movement of the robotic fish implementing the thunniform principle of locomotion. The shape of the body and the tail fin of the robotic fish were designed using a computational model simulating a virtual body in an aquatic medium. A prototype of a robotic fish was constructed and tested in experimental conditions. Dependencies of fish velocity on the dynamic characteristics of tail oscillations were analyzed. In particular, it was found that the robot’s speed increased as the frequency of tail fin oscillations grew. We also found that for fixed frequencies, an increase in the oscillation amplitude lead to an increase in the swimming speed only up to a certain threshold. Further growth of the oscillation amplitude lead to a weak increase in speed at higher energy costs.

## 1. Introduction

There have been multiple attempts to develop a biomimetic robot. Such robots intend to reproduce principles of movement of living creatures in nature. These principles have been continuously perfected by nature to ensure the survival of creatures. Such evolutionary modifications took millions of years of adaptive interaction with the external environment. In technical terms, survival is provided by two obvious fundamental points. The first one concerns minimizing energy consumption during movement. The second one refers to maximizing movement efficacy; e.g., speed or distance traveled, or acceleration, in conditions with limited energetic resources. Another important aspect of survival is the way creatures interact with the natural environment. Specifically, movement should be implemented in such a way that possible perturbations of physical parameters of air or aquatic environment are reduced to a minimum. By optimizing certain functions, the evolutionary process fosters biodiversity. In animals living in the water, diversity is manifested in different types of swimming with different characteristics of body movement. Designing a creature-like biomorphic machine represents a very attractive, but very complicated engineering task. To date, quite a lot of robotic fishes with different types of swimming and different bioinspired propulsive mechanisms have been proposed (for recent reviews, see, e.g., [1,2,3]).

Tunas represent a group of fishes with many evolutionary adaptations (e.g., tail shape, lateral peduncle keels, pectoral fin shape, finlets, etc.) making them high-performance swimmers. Because of its effect on the speed and efficiency of swimming, the thunniform method of locomotion attracted researchers and inspired numerous biomimetic underwater robots. For thunniform locomotion, the undulation is limited typically to the rear one-third of the body and reaches the maximal amplitude at the end of the tail peduncle [4,5]. The oscillatory tail fin movement generates thrust and yields robot translational motion. Thunniform swimming is considered as a primary locomotion mode for many fast-moving robotic swimmers. The first tuna-mimetic robot, RoboTuna, was built at MIT [6,7]. Later, the vorticity control unmanned undersea vehicle (VCUUV) was developed based on RoboTuna, with some improvements and additional capabilities, such as obstacle avoidance and the use of up-down motion [8]. In contrast to the classic multi-joint robot layout, a robotic platform with a compliant body was proposed in [9]. The prototype driven by a single actuator featured an anisotropic material distribution along its body length to provide the desired swimming motions. Further evolution of the flexible housings approach was presented in [10], which provided design samples for different fish-like robots (including robotic tuna) and actuation units. Another elegant technical solution was implemented in the CasiTuna robot [11]. Unlike most multiple-link propulsive mechanisms used in robotic fish (where each joint contains an actuator, e.g., [12]), this version had two-motor-actuated motors placed in the anterior body, which allowed it to achieve tuna-like lateral undulations. Masoomi et al. [13] developed a tuna-like robot called UC-Ika 1 with a flexible body to investigate and improve the swimming performance of the biomimetic swimming robots. Electromagnetic-driven multi-joint bionic fish design and different control strategies were proposed in [14]. The presented approach allowed for high-frequency (up to 5 Hz) body undulation. Tunabot developed by Zhu et al. [15] also focused on high-frequency flapping (up to 15 Hz) to improve swimming speed and energy efficiency. White et al. [3] improved the energy efficiency and swimming performance of Tuna Flex through multiple passive joints. Combining the advantages of single-joint and multijoint robotic fish, a novel mechanism of redundant joints was presented in [16] to enrich the swimming patterns and maneuverability of the robotic tuna.

Thus, the central task of designing fish-like robots is to find a solution that provides, along with speed characteristics, high energy efficiency. Unlike real fish that use their efficient muscles to realize both fast swimming and steering, the robotic fish has to employ a combination of several motors. Another tuna robot was designed using shape memory alloy (SMA) wires as muscles [17]. When SMA wires are heated, they contract and when cooled, they expand. This driving mechanism in the robotic fish allowed the model to achieve a measurable thrust. However, it required large amounts of power for operation. The authors found this solution to be energetically more costly than the traditional use of an electric motor as a source of tail oscillations. An alternative approach, which is gaining popularity, is to use the wire-driven mover, which also imitates muscle contractions. There have been several papers describing underwater robots with subcarangiform and carangiform locomotion and various motor layout options [18,19,20,21,22,23,24]. These robots consist of rigid elements and elastic cables that represent the bone/tissue and muscles of real fish. The body stiffness of the robot can be adjusted to approximate the same stiffness as a real fish and modulate its performance characteristics. The strategic aim for further research is to reach real animal efficiency in the robotic propulsion system. To do this, it is necessary to study and implement different aspects of the movement of fish in robots. For example, it has been recently shown that body flexibility and stiffness can significantly influence swimming speed [25].

Biomimetic robotic fishes with wire-driven movers demonstrate impressive results in movement speeds, reaching up to 2 body lengths per second (BL/s) and higher [18,20]. However, they are quite complicated and expensive to manufacture. The aim of our work was to develop a simplified, low-cost, but bio-inspired propulsion system and a robotic fish based on this system. We deliberately simplified the tail section of the robot. Such a solution could increase drag by increasing turbulent flows. On the other hand, this negative effect could be compensated by the increased energy efficiency due to the reduction of mass and rigidity of the body part bent by the servo drive. We investigated a computational model of the robot in the aquatic environment and performed physical experiments with the robot swimming in a water pool to analyze the influence of dynamic parameters (amplitude and tail beat frequency) of the propulsion system on the robot’s speed and efficiency.

## 2. Robotic Fish Design

Figure 1 illustrates the design of our robotic fish. The nose of the robot’s body was scaled from the real geometry of a tuna fish. Based on the photographs, a 3D model was built. It was then adjusted to fit all the necessary equipment. During the development, the NX computer-aided design system was used. The tail section of the robot was deliberately simplified. Such a solution can increase drag by increasing turbulent flows. On the other hand, this negative effect can be compensated by the increased energy efficiency due to the reduction of mass and rigidity of the body part bent by the servo drive.

The robot body housed the JX-DC5821LV servo drive, which was additionally sealed. The power consumption rating of this servo was 6 V and approximately 1 A. Moreover, ballast in the form of lead weights was installed in the hull to compensate for uneven weight distribution. Polystyrene was added to reach zero buoyancy. During the operation, the inside of the body was filled with water. The mass of the robot filled with water and having zero buoyancy was 0.78 kg.

The propulsion system was located in the back part of the robot and is shown in detail in Figure 2A. It consisted of a servomotor fixed on the robot’s body, flexible plate, and tail fin. Both sides of the flexible plate had rods that deformed it when the servo rotates. To give an additional degree of freedom, the tail fin was mounted on a spring-loaded hinge (see Figure 2B). The parameters of the flexible plate were 114 m × 0.4 mm × 13 mm. The plate was made of steel cold-rolled heat-treated tape. The Young’s modulus according to the measurement results was 195 GPa.

Tail oscillations were provided by two wire rods playing the roles of flexors and extensors mimicking muscular contractions. The resulting robotic fish demonstrated qualitatively thunniform movement when the tail oscillations were converted into translational motion of the whole robot. In our model, the control system can gradually change the amplitude and frequency of tail oscillations.

In computer simulations, we also tested the robot’s movement in an aquatic environment using ANSYS software. The main goal of optimizing the movement of a robotic fish was to achieve the best possible relationship between tail geometry and the power consumption of the drive, which was the basic driving force. A prototype was developed to verify the computer model of a biomorphic mover, which can simulate the movement of fish. Calculations included several stages: building a computer model from different computational modules presented inside the program, loading the studied geometry (3D model), building a computational grid on the surface of the geometry under study, setting the initial conditions, carrying out calculations, and analyzing the results obtained. Specifically, we looked at the hydrodynamics of the body resistance arising in the fluid flow using our 3D model. The results are shown in Figure 3.

## 3. Fish Swimming Experiments

For physical experiments with the robotic fish, we developed the experimental setup shown in Figure 4. The setup was capable of registering the kinematics of swimming. It consisted of a carriage with a camera attached to it, which moved along rail guides. The camera captured the movements of the robot against the background of the coordinate grid, which then made it possible to further digitize the movements.

First, it was necessary to move from the control parameters of the servo to the real parameters of the tail beat. To do this, in the first part of the experiment, a series of servo starts were repeated with the infixed position of the body, with a video recording against the background of the coordinate grid. The actual values of frequency and amplitude of the tail oscillations were determined in the video material for different operating modes.

Then, we detected the maximal performance of the robotic fish. The moving speed was limited by the maximal parameters of the servo drive and the composition of the tail part. Specifically, the maximum speed was about 0.4 body lengths per second (BL/s), which was achieved at the maximal beat frequency of 3.4 Hz, with the maximal tail deflection amplitude of 105 mm. Here, we also chose a minimum speed of 0.1 BL/s, below which the experiments were not carried out. With an arbitrarily chosen minimum amplitude of 30 mm, the minimum frequency was 1.9 Hz.

For the main part of the experiments, we divided the resulting tail beat frequency range into three modes: 3.4 Hz, 2.5 Hz, and 1.9 Hz. The amplitude range of 30–105 mm was divided into a large number of points for further analysis of the dependence of performance on the tail beat amplitude.

The robot was placed in the pool (Figure 4A); then, a control command was sent from the control panel (Figure 4B) by wire. The control panel was constantly located above the fish, which, combined with the use of flexible and thin wires, minimized the external impact on the movement of the robot. The control command triggered the movements of the robot with a given frequency and amplitude of tail oscillations. The robot’s swimming was recorded on video.

The experimental protocol was as follows. A series of launches of the robot was carried out at different frequencies and amplitudes of tail flaps. Before each launch, the parameters of the amplitude and speed were set for the controller. These parameters did not change throughout each launch. The robot was moving in the water column, over a marked grid with a cell spacing of 100 × 100 mm. While the robot was moving, filming was carried out on a video camera with further processing of images on the computer. A segment of the path was selected, within which the robot’s speed was constant (after launch, it accelerated until it reached a constant speed); the length of the distance traveled (according to the markings) and the time it took for the robot to swim along this path was determined. Then, the speed of translational movement was calculated (not taking into account the rocking of the robot from the flapping of the tail). A number of launches were carried out with different control parameters and the velocities were calculated for each resulting mode. Ten repetitions were carried out for each set of parameters.

Figure 5 shows the sequence of snapshots of the robot swimming in the water pool compared to a real tuna. In addition, a typical video of the swimming robot is provided in the Appendix A. Just like in real tuna, the robotic fish can have a point of inflection in the midline at the point of attachment of the caudal fin, although less pronounced in the case of the robot. A spring-loaded hinge (Figure 2A) is responsible for this inflection, connecting the tail with an elastic plate.

The results illustrating the dependences of the actual robot speed on the tail oscillation amplitude and frequency are presented in Figure 6. One can note that the data sets have non-monotonic shapes with a seemingly oscillatory pattern. Results for different frequencies have qualitatively similar profiles, which shifted to a higher speed as the tail beat frequency increased. We notice that all dependencies of the robot speed on the amplitude of the tail beats have two linear trends with different slopes. Results of linear regression Speed=a×Amplitude+b show a = [0.005, 0.008] for a small amplitude value (Figure 6, solid lines) and a≈ 0.001 for a larger value (Figure 6, dashed lines). The intersection of the regression lines takes place at an amplitude of about 50 mm for all the beat frequencies.

We hypothesized that the oscillatory character of the dependences may be a result of a resonant interaction of an oscillating body and hydrodynamic waves reflected from the walls of the pool. In other words, during the movement, the water in the pool forms a two-dimensional profile of a standing wave defined mostly by the pool geometry. Our robotic fish in this case moves over “the non-flat landscape grid” modulated by such an interaction. As one may expect, the characteristics of such movement can be quite sensitive to each particular trial condition (the vertical bars in Figure 6 show the standard deviation, *n* = 10).

We quantify energy efficiency in terms of the cost of transport (COT) for comparison among different swimming modes. COT, which describes the energy required to move a unit mass over a unit speed, is calculated with the following expression:(1)$$\mathrm{COT}=\frac{P}{v\cdot m}\;,\;$$
where *P* (W) is the total electrical power drawn by the motor, *v* (m/s) is the swimming speed, and *m* (kg) is the robot mass (0.780 kg). COT has the unit of J kg^−1^ m^−1^.

Figure 7 shows the dependence of the cost of transport on the amplitude of the tail beats. For all three frequencies, we can observe a minimum COT corresponding to the change in the angle of the regression lines in Figure 6. Summarizing the results (Figure 6 and Figure 7), we can conclude that the speed of the robot strongly depends on the amplitude of the tail beats up to a threshold value of about 50 mm. With a further increase in the amplitude, the robot can swim a little faster; however, this mode is more energy expensive.

More spectacular is the presentation of the COT versus robot movement speed shown in Figure 8. For higher frequency values, there is a threshold speed, below which the swimming is efficient. The threshold speed is higher for larger frequency values (marked areas in Figure 8). Note that for lower frequencies, the movement speed is limited and does not increase with increasing COT, i.e., the oscillation amplitude (black labels in Figure 8).

Summarizing the experimental results, we discovered that the maximum robot speed with the minimum COT is observed at a high frequency of a tail beat and a rather small amplitude. With an increase in the amplitude above a certain threshold (about 50 mm), the body starts to sway around the longitudinal axis, which reduces the efficiency of the translational movement. With a different shape of the body, for example, as in fish with a flattened body elongated along the vertical axis or a large area of fins located in a vertical plane, this parasitic swaying would not occur. However, for the shape characteristic of tuna, flapping with a high frequency and low amplitude shows the greatest efficiency.

## 4. Conclusions

We developed a thunniform swimming robot imitating tuna fish movement. We proposed a novel simplified propulsion system composed of an elastic cord with flexor/extensor mechanical systems mimicking muscular contraction in living animals. This system provides oscillatory movement of the tail fin that, in turn, generates translational motion of the robot. The shape and geometry of the robot body mimicking tuna parameters were first designed in a computer model and subsequently, implemented in a working robotic device.

The maximum speed of our robotic fish was about 0.4 BL/s, which is below the range of 0.58–2.15 BL/s shown by robots with a wire-driven mover without tail simplification [18,19,20,22,23]. However, the presented robot demonstrated a speed higher than in other works (0.22 BL/s [26], 0.254 BL/s [27], 0.308 BL/s [28]), where a simplified version of the tail section was also used. Obviously, the advantages of a simplified design (the reduction of mass and rigidity of the body part bent by the servo drive) still cannot compensate for the increased drag caused by elevated turbulent flows. The loss in speed in this case can be considered a price for the simplicity and low cost of the robot.

A series of experiments investigating how the robot’s kinematics depends on the dynamic parameters of the propulsion system were carried out. It was found that the robot’s speed increased as the frequency of tail fin oscillations grew. We also found that for fixed frequencies, there is an interval of energetically preferable traveling speeds up to a threshold speed. Movement with a higher speed was also possible; however, it appeared to be more power-consuming. This conclusion qualitatively agreed with the data of COT studies from living tunas [4,29,30], although quantitatively, the values obtained for the robot were still higher. Thus, the swimming efficiency of live tuna lies in the interval 1–7 J kg^−1^·m^−1^ depending on the swimming speed. The values obtained for the presented robot were 30–70 J kg^−1^·m^−1^ for different values of frequency and swimming speed. The values obtained were also higher compared to the most outstanding results in this area: 4.06–11.80 J kg^−1^·m^−1^ [16] and 4.5–15.0 J kg^−1^·m^−1^ [3].

Note that the paper studied the dependence of the robot swimming efficiency on amplitude in addition to works that considered the dependence on tail beat frequency, or dependence of frequency and swimming speed [3,14,15,18]. We found that for the propulsion system presented (at a fixed frequency), an increase in the oscillation amplitude only up to a certain threshold leads to an increase in the swimming speed. A further increase in the oscillation amplitude leads to a weak increase in speed at higher energy costs. The nature of this dependence is apparently related to the conditions of the experiment. A two-dimensional profile of a standing wave is formed due to the wave reflection from the walls of the pool, which affects the speed of the robot. In addition, an assessment of the energy efficiency was performed in dependence on the dynamic parameters of the tail oscillation. It was shown that with an increase in the tail oscillations amplitude above the threshold, the cost of transport increases. We also found that for fixed frequencies, there is an interval of energetically preferable traveling speeds up to a threshold speed. Movement at a higher speed is possible; however, it is more power-consuming. Generally, to increase the swimming speed, it is preferable to increase the oscillation frequency of the caudal fin, rather than the amplitude. These conclusions are in qualitative agreement with the results of the numerical simulation of thunniform swimming [31,32].

We note that experimental results with general tendencies in decreasing dependencies (see, for example, Figure 6) have quite a high degree of variability. This is a result of problem complexity—a 3D-body dynamics in the hydrodynamical medium can acquire additional degrees of freedom competing with translational motion and consuming energy. To overcome these problems, dynamic feedback systems should be further developed to compensate for such physical fluctuations. Note that in recent studies, such dynamic adjustments of the cord stiffness have led to a significant increase in swimming performance [7]. We also plan to include adaptive feedback into the control systems driving the tail oscillations that would compensate for non-translational fluctuations and dynamically converge to an optimal swimming mode.

## Figures and Tables

**Figure 1 biomimetics-07-00215-f001:**
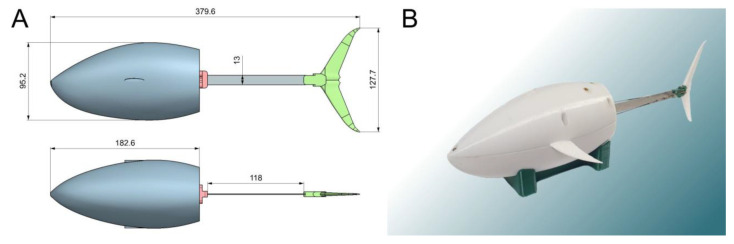
The robotic fish. (**A**) Dimensional drawing; (**B**) working prototype.

**Figure 2 biomimetics-07-00215-f002:**
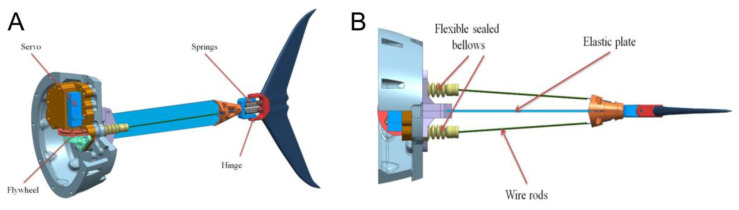
Robotic fish propulsion system. (**A**) The tail fin is connected to a servomotor; (**B**) muscular ligaments-like wire provides flexor/extensor movement of the tail fin mounted to the elastic rod.

**Figure 3 biomimetics-07-00215-f003:**
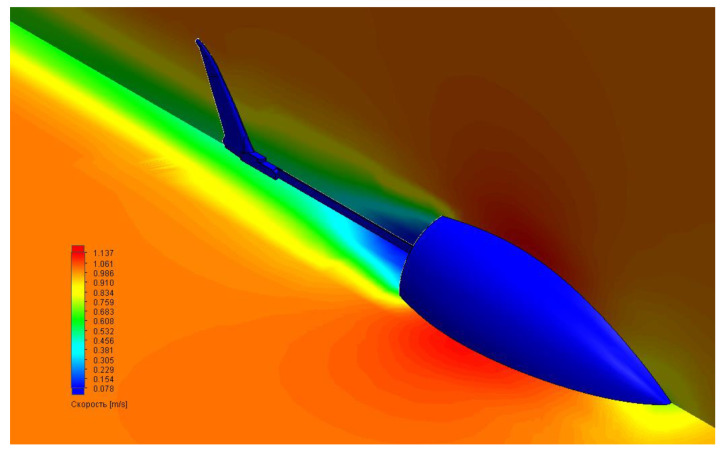
Simulation of body resistance profile of the 3D fish robot movement in an aquatic medium with constant flow. Color grade corresponds to values of the flow speed.

**Figure 4 biomimetics-07-00215-f004:**
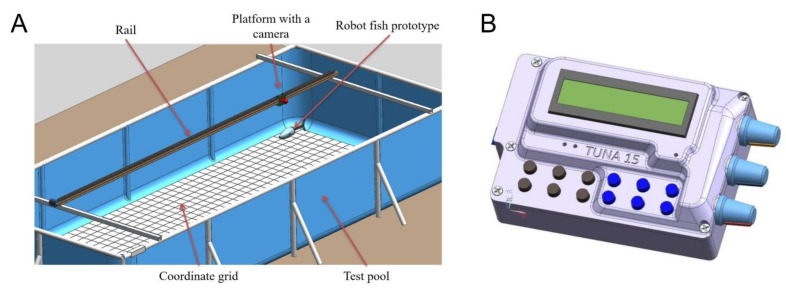
Fish swimming experiments. (**A**) Experimental setup to monitor characteristics (speed, direction, mode of movement) of the robotic fish. (**B**) Control panel of the robotic fish.

**Figure 5 biomimetics-07-00215-f005:**
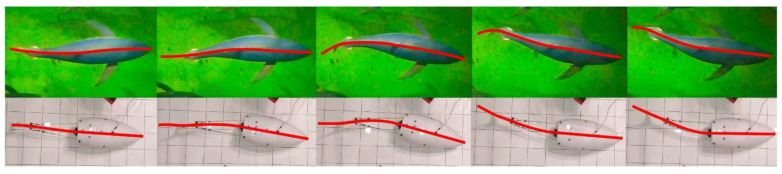
Snapshots from a video of a tuna and the robotic fish swimming. The middle line is marked in red.

**Figure 6 biomimetics-07-00215-f006:**
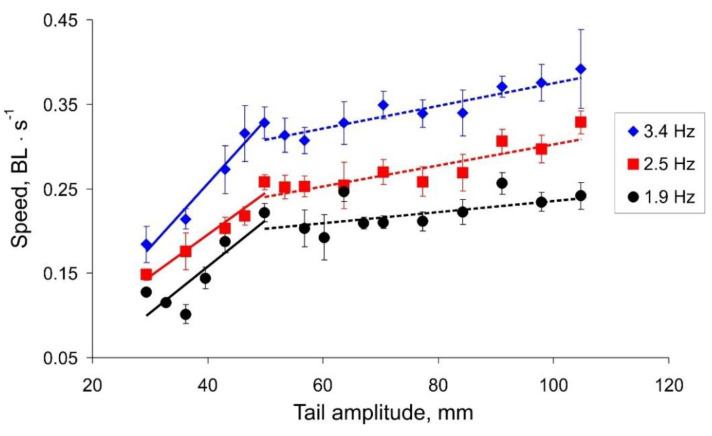
The robot speed vs. the tail amplitude in three modes of the tail beat frequency.

**Figure 7 biomimetics-07-00215-f007:**
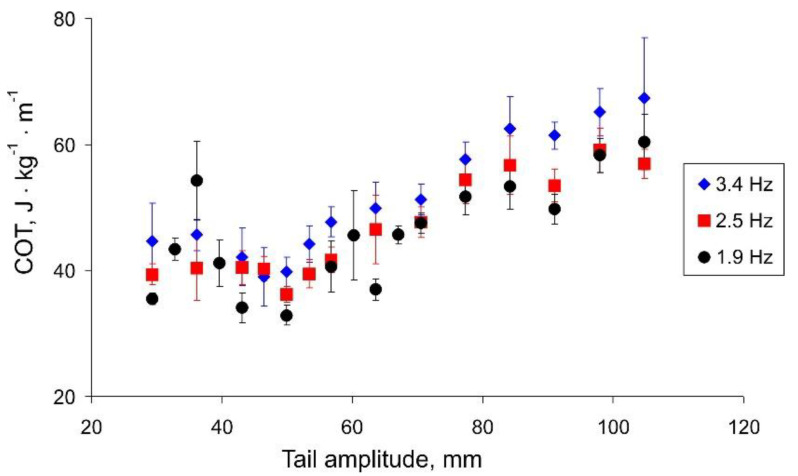
The cost of transport (COT) vs. the tail amplitude.

**Figure 8 biomimetics-07-00215-f008:**
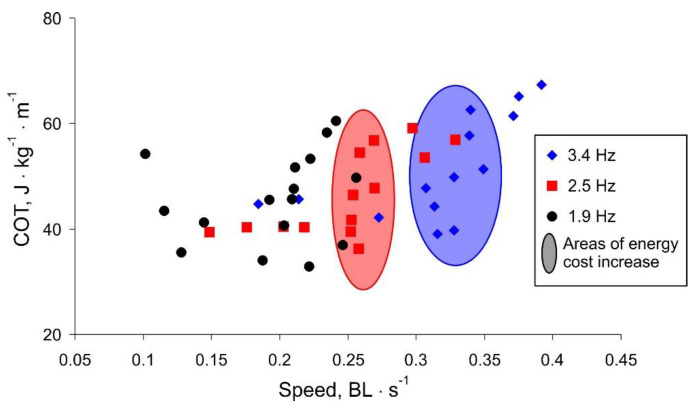
Summary characteristic of the robot movement. Marked areas illustrate a threshold speed above which the increase of movement speed is energetically expensive.

## Data Availability

Not applicable.

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
