# Peer review of "Bioinspired Propulsion System for a Thunniform Robotic Fish"

_biomimetics, 2022, doi:10.3390/biomimetics7040215_

Round 1
Reviewer 1 Report
This paper proposed a biomorphic propulsion system for underwater robotic fish, which move forward by the oscillatory tail and implement thunniform locomotion. In detail, the tail fin is driven by servomotor and simulating muscle contraction by two symmetric movable thrusts. The prototype of the robot device is constructed, and the tail oscillations with various amplitude and frequency values are tested to analyze the robot’s velocity. The paper is complete and self-consistent, but some issues are still worth paying attention to.
1. The electromechanical parameters of the system are advised to be added.
2. The proposed tail mechanism is novel, but the experiments on amplitude and frequency are common. More discussions on novel structures and scientific innovations are expected.
3. The syntax and figure numbering may be incorrect in detail. For example, the citation “Figure 7” in line 169 of page 5 may be mistakenly written as “Figure 8”.
4. The argument “in case of fixed frequency, the raise of oscillation amplitude, may not correspond to a higher swimming speed.” in line 217 of section 4 does not seem to be able to be evidenced from the experimental data in Figure 7. More references can be added to prove the description.
Author Response
We thank the reviewer for comments helping to improve the manuscript. For convenience, all new parts of the text and rewritten paragraphs have been marked blue.
- The electromechanical parameters of the system are advised to be added.
The parameters of robotic fish were added to the manuscript.
- The proposed tail mechanism is novel, but the experiments on amplitude and frequency are common. More discussions on novel structures and scientific innovations are expected.
Indeed, the study of the speed of the robot and its efficiency as a function of the tail beat frequency has been investigated by many researchers mostly in simulations. We added references to those works. We worked with real prototype and added some more discussions concerning this issue.
- The syntax and figure numbering may be incorrect in detail. For example, the citation “Figure 7” in line 169 of page 5 may be mistakenly written as “Figure 8”.
Corrected.
- The argument “in case of fixed frequency, the raise of oscillation amplitude, may not correspond to a higher swimming speed.” in line 217 of section 4 does not seem to be able to be evidenced from the experimental data in Figure 7. More references can be added to prove the description.
We agree. Corrected in the revised version.
Reviewer 2 Report
In this study the authors present a tuna-inspired robot with a tuna-like tail fin and anterior body, and a non-tuna-like posterior body that is oscillated by elastic cords and servomotors. I think this paper could be an interesting contribution but it is currently missing some details that make it difficult to evaluate. My major suggestions are to add details throughout the manuscript about how the robot was built (was it always tested without a posterior body cover? How large is it? What is the stiffness of the elastic plate? Is it tethered? Etc. etc.), how it was tested (how did you control kinematics and what exactly are the kinematic parameters you looked across? What are the maximum ranges of kinematics for this robot? Provide measurements in units that are more common in this field), and what was measured (are you measuring tail oscillation or is it prescribed during control? How many trials did you run for each set of kinematics? Etc). Some more information is needed throughout the manuscript to understand what you have done here and to assess the value of your work.
Similarly, I think more details are needed to evaluate how well this robot mimics a tuna. The easiest way I can think of is comparing some aspect of tuna swimming kinematics (typically from the dorsal midline) to the kinematics of the robot. As of now, it just seems like you made a robot that is shaped a little like a tuna, but I don’t know that it is performing like a tuna. There are a few studies with published tuna swimming kinematics which could be compared to your robot to see how similar they are.
It would also be good to add a little more justification for why tunas are a good group to choose for inspiration in designing a robot. E.g. we think of them as high-performance swimmers that can swim at high speeds (although we don’t necessarily think of tunas as efficient). Tunas also have many modifications that we think help them be high-performance, such as tail shape, fairings, pectoral fin shape and pectoral fin base structure, finlets, lateral peduncle keels, etc.).
You make a point in the introduction of saying that we need to think about designing robots that are energy efficient. And while you produce some data on efficiency, it’s hard to know if these are good efficiencies or not compared to other fish-inspired robots, or compared to biology. Can you find some comparable data in the literature on both fishes and robots to compare your values to and discuss it in your paper? In other words, how efficient is your robot compared to other tuna-like or fish-like robots? How efficient is your robot compared to real tunas or fishes? (Also, a measurement of power efficiency might be more comparable to other data in the literature on robots and fishes).
I feel that somewhere in this manuscript you need to mention and discuss Strouhal number as it is relevant when thinking about tail beat kinematics and efficiency.
Below are also a list of suggestions/edits in the order that they appear in your manuscript. Please consider making the changes below as I believe they would improve your manuscript and make it more clear for readers.
Title and throughout: I would generally recommend that you use the term “bioinspired” instead of “biomorphic” as I believe it is a more common term. You could also use the term “biomimetic” as well, although the meaning is slightly different…e.g. (biomimetic) mimicking biology vs (bioinspired) inspired by biology.
Line 23: “These principles have been optimized…” This is not always true. Evolution is not an optimization force for single functions and it has no preconceived direction (e.g. fishes today are not faster than fishes from a million years ago…in fact there is a lot of diversity of swimming behavior among fishes that are alive today). In reality, biological structures and behaviors just need to work well-enough for organisms to survive and reproduce, and in many cases biological structures or behaviors will have many functions and cannot be optimized for a single function because of this. Consider using a different phrase instead of “optimized” – this is a minor point but an important one because of how this misconception can lead to incorrect assumptions about evolution and optimization.
Lines 64-65: “Also, it was recently shown that body flexibility 64 and stiffness can significantly influence on swimming speed [15]” This sentence doesn’t connect well with the ideas of the previous and following sentences – consider editing it to make your connections and thoughts more clear or moving it to a more appropriate place in the manuscript.
Line 67: Around here it might be good to add another sentence or two that describes both the benefits and drawbacks of using the “wire-drive mover” as an actuating device.
Figure 1 and Lines 77-90: You need to provide some basic dimensional information about your robotic tuna. How long is it? How tall is the tail and body? Consider adding lateral view and dorsal view photographs of the robot with scale bars to Figure 1…currently Figure 1B looks nice but there is no scale bar and the view is tilted so it would be hard to determine any dimensions from this image. Dimensional information about the robot needs to be added and made more accessible in your manuscript.
Lines 81-84: Is the robot free-swimming or untethered? It doesn’t look like it based on Figure 5, but please be clear about this in the text somewhere here.
Figure 1B: Is the robotic model just the elastic plate and wire rods for the posterior body as shown photographed in this figure and in Figure 5? Or is the posterior body more covered like in fig 1A? Please be clear about this in the text.
Line 135-145 – Somewhere here you should list the kinematic parameters or ranges of parameters that you tested at. These numbers should not just show up in the graphs, but should also be in the text for clarity. And then you should justify why you decided to use those ranges and values? E.g. why did you use those three tail-beat frequency values? Are they biologically relevant? And same with the different amplitude/oscillation values?
Line 136-137 and Figure 6: Is the “amplitude of tail flaps” at this line and the “tail oscillation angle” in Figure 6 the same thing? If so, it is common to measure amplitude either as a distance (e.g. cm) or as a percentage of body length. Consider providing it in more familiar and comparable units if so.
Figure 6 and lines 151-157: I don’t understand what tail oscillation angle is and why it is changing at different tail-beat frequencies. Do you have active control over just the tail angle somehow? Is amplitude of the oscillation also changing for these different motion parameters? (if so you should say so and you probably also need to measure and present the changes here). You need to add a more detail here to help the readers understand what tail oscillation angle is (how you are measuring this parameter) and how mechanically it is able to change at a give tailbeat frequency. I’m also just guessing that the three modes (given in Hz values) are different tailbeat frequencies? Please be clear about this as well.
Figure 6: I would suggest presenting movement speed as body-lengths per second either instead of meters per second or as a second graph in addition. Body lengths per second is generally more informative for comparing among robots or fishes. And if you provide the length of the robot’s body length in the text, it is easy to calculate m/s.
Figures 6, 8, and 9: Can you add error bars to these plots?
Figure 7 and Line 160: What were the experimental parameters and how did you decide on them? You need to provide some more information here for readers to understand what you did in you experiment.
Lines 169-172: You mention figure 8 and how it has vertical bars, but I don’t see any vertical bars in Figure 8. Perhaps you mean to say Figure 7?
Lines 175-185: It might be better and more comparable (to other papers) if you can also (or instead) measure power consumption of the servomotors and use that to calculate efficiency. Also, shouldn’t efficiency be dimensionless? Maybe this term you are using should have a different or more specific name.
Line 187: is the tail oscillation amplitude you mention here the same as the alpha parameter from equation 2? (aka tail oscillation angle). You need to provide a little more info to help readers understand what this (or these parameters) mean and how they are being measured.
Line 209: you say that your robot imitates tuna fish movement, but I don’t see any evidence provided for this. For example, you could compare the midline kinematics (or some other relevant parameter) of your robot to midline kinematics for tunas, but I think more evidence is needed for you to make this claim. As of now I can also see that the front half of the body of your robot is based on a tuna.
Consider providing a video, some screenshots, and data for midline kinematics of a robot to help readers better understand what the robot looks like and how its motion compares to real tunas/fishes.
References – check the formatting of your references as there are a couple of minor errors there.
Author Response
We thank the reviewer for careful reading of the manuscript and deep questions that helped us improve the manuscript. We have restructured and thoroughly rewritten some part of the manuscript. In particular, we added details about design and testing of the robotic fish. For convenience, all new parts of the text and rewritten paragraphs have been marked blue.
Below, please find a point-by-point account for questions and critical comments.
Similarly, I think more details are needed to evaluate how well this robot mimics a tuna. The easiest way I can think of is comparing some aspect of tuna swimming kinematics (typically from the dorsal midline) to the kinematics of the robot. As of now, it just seems like you made a robot that is shaped a little like a tuna, but I don’t know that it is performing like a tuna. There are a few studies with published tuna swimming kinematics which could be compared to your robot to see how similar they are.
We agree. Indeed, the shape was taken from real tuna, while the propulsion system may be much simpler. Actually, we did not claim to make an accurate model of real fish, it is much more complicated task. However, our propulsion system makes movements similar (of course, qualitatively) to what real fish does. To be more convincing we've included snapshots of the movement of a real tuna vs. robot and marked a dorsal midline. In addition, a typical video of the floating robot is provided in the supplementary materials.
In case the video doesn't load on submission:
https://drive.google.com/file/d/1anDP3uuzOnN39BmpAuxaZWQMahJGUKXv/view
It would also be good to add a little more justification for why tunas are a good group to choose for inspiration in designing a robot. E.g. we think of them as high-performance swimmers that can swim at high speeds (although we don’t necessarily think of tunas as efficient). Tunas also have many modifications that we think help them be high-performance, such as tail shape, fairings, pectoral fin shape and pectoral fin base structure, finlets, lateral peduncle keels, etc.).
We agree with the reviewer. Corrections to the introduction have been made.
You make a point in the introduction of saying that we need to think about designing robots that are energy efficient. And while you produce some data on efficiency, it’s hard to know if these are good efficiencies or not compared to other fish-inspired robots, or compared to biology. Can you find some comparable data in the literature on both fishes and robots to compare your values to and discuss it in your paper? In other words, how efficient is your robot compared to other tuna-like or fish-like robots? How efficient is your robot compared to real tunas or fishes? (Also, a measurement of power efficiency might be more comparable to other data in the literature on robots and fishes).
We agree with the reviewer. Comparison of the robot efficiency relative to live tunas and the most outstanding robots have been added to the conclusion.
I feel that somewhere in this manuscript you need to mention and discuss Strouhal number as it is relevant when thinking about tail beat kinematics and efficiency.
In the revised manuscript we used cost of transport (COT) to analyze robots’s kinematics and efficiency.
Below are also a list of suggestions/edits in the order that they appear in your manuscript. Please consider making the changes below as I believe they would improve your manuscript and make it more clear for readers.
Title and throughout: I would generally recommend that you use the term “bioinspired” instead of “biomorphic” as I believe it is a more common term. You could also use the term “biomimetic” as well, although the meaning is slightly different…e.g. (biomimetic) mimicking biology vs (bioinspired) inspired by biology.
We agree. Following the suggestions the title was corrected.
Line 23: “These principles have been optimized…” This is not always true. Evolution is not an optimization force for single functions and it has no preconceived direction (e.g. fishes today are not faster than fishes from a million years ago…in fact there is a lot of diversity of swimming behavior among fishes that are alive today). In reality, biological structures and behaviors just need to work well-enough for organisms to survive and reproduce, and in many cases biological structures or behaviors will have many functions and cannot be optimized for a single function because of this. Consider using a different phrase instead of “optimized” – this is a minor point but an important one because of how this misconception can lead to incorrect assumptions about evolution and optimization.
We agree. Corrected.
Lines 64-65: “Also, it was recently shown that body flexibility 64 and stiffness can significantly influence on swimming speed [15]” This sentence doesn’t connect well with the ideas of the previous and following sentences – consider editing it to make your connections and thoughts more clear or moving it to a more appropriate place in the manuscript.
We corrected the introduction, including taking into account this remark.
Line 67: Around here it might be good to add another sentence or two that describes both the benefits and drawbacks of using the “wire-drive mover” as an actuating device.
We agree with the reviewer. A brief description of this type of robot was added in the introduction.
Figure 1 and Lines 77-90: You need to provide some basic dimensional information about your robotic tuna. How long is it? How tall is the tail and body? Consider adding lateral view and dorsal view photographs of the robot with scale bars to Figure 1…currently Figure 1B looks nice but there is no scale bar and the view is tilted so it would be hard to determine any dimensions from this image. Dimensional information about the robot needs to be added and made more accessible in your manuscript.
We agree. We revised Figure 1 to take into account the comments.
Lines 81-84: Is the robot free-swimming or untethered? It doesn’t look like it based on Figure 5, but please be clear about this in the text somewhere here.
We added description concerning this issue in section 3. Specifically the control panel was constantly located above the fish, which, combined with the use of flexible and thin wires, minimized the external impact on the movement of the robot.
Figure 1B: Is the robotic model just the elastic plate and wire rods for the posterior body as shown photographed in this figure and in Figure 5? Or is the posterior body more covered like in fig 1A? Please be clear about this in the text.
We corrected the figures and text according to the comment.
Line 135-145 – Somewhere here you should list the kinematic parameters or ranges of parameters that you tested at. These numbers should not just show up in the graphs, but should also be in the text for clarity. And then you should justify why you decided to use those ranges and values? E.g. why did you use those three tail-beat frequency values? Are they biologically relevant? And same with the different amplitude/oscillation values?
We added a new part in the section 3 concerning ranges of the kinematic parameters.
Line 136-137 and Figure 6: Is the “amplitude of tail flaps” at this line and the “tail oscillation angle” in Figure 6 the same thing? If so, it is common to measure amplitude either as a distance (e.g. cm) or as a percentage of body length. Consider providing it in more familiar and comparable units if so.
Corrected. In new version of the manuscript we use the amplitude (mm) only.
Figure 6 and lines 151-157: I don’t understand what tail oscillation angle is and why it is changing at different tail-beat frequencies. Do you have active control over just the tail angle somehow? Is amplitude of the oscillation also changing for these different motion parameters? (if so you should say so and you probably also need to measure and present the changes here). You need to add a more detail here to help the readers understand what tail oscillation angle is (how you are measuring this parameter) and how mechanically it is able to change at a give tailbeat frequency. I’m also just guessing that the three modes (given in Hz values) are different tailbeat frequencies? Please be clear about this as well.
We added a new part in the section 3 concerning ranges of the kinematic parameters. No active control was used.
Figure 6: I would suggest presenting movement speed as body-lengths per second either instead of meters per second or as a second graph in addition. Body lengths per second is generally more informative for comparing among robots or fishes. And if you provide the length of the robot’s body length in the text, it is easy to calculate m/s.
Corrected
Figures 6, 8, and 9: Can you add error bars to these plots?
Corrected
Figure 7 and Line 160: What were the experimental parameters and how did you decide on them? You need to provide some more information here for readers to understand what you did in you experiment.
We added a new part in the section 3 concerning the experimental parameters.
Lines 169-172: You mention figure 8 and how it has vertical bars, but I don’t see any vertical bars in Figure 8. Perhaps you mean to say Figure 7?
Corrected.
Lines 175-185: It might be better and more comparable (to other papers) if you can also (or instead) measure power consumption of the servomotors and use that to calculate efficiency. Also, shouldn’t efficiency be dimensionless? Maybe this term you are using should have a different or more specific name.
In the new version of the manuscript we use COT to analyze robots’s kinematics and efficiency.
Line 187: is the tail oscillation amplitude you mention here the same as the alpha parameter from equation 2? (aka tail oscillation angle). You need to provide a little more info to help readers understand what this (or these parameters) mean and how they are being measured.
In the revised version of the manuscript equation 2 was not used. Instead we used COT to analyze robots’s kinematics and efficiency.
Line 209: you say that your robot imitates tuna fish movement, but I don’t see any evidence provided for this. For example, you could compare the midline kinematics (or some other relevant parameter) of your robot to midline kinematics for tunas, but I think more evidence is needed for you to make this claim. As of now I can also see that the front half of the body of your robot is based on a tuna.
Consider providing a video, some screenshots, and data for midline kinematics of a robot to help readers better understand what the robot looks like and how its motion compares to real tunas/fishes.
As suggested by the reviewer, we've included snapshots of the movement of a real tuna vs. robot and marked a dorsal midline. In addition, a typical video of the floating robot is provided in the supplementary materials.
In case the video doesn't load on submission:
https://drive.google.com/file/d/1anDP3uuzOnN39BmpAuxaZWQMahJGUKXv/view
References – check the formatting of your references as there are a couple of minor errors there.
Corrected.
Reviewer 3 Report
The authors presented a tuna-like robotic fish with a wire-driven propulsion system mounted on an elastic plate. However, its swimming performance is not very impressive.
1.Since the dimensions of the robotic fish are not presented in this paper, it is very difficult to evaluate the swimming performance. Additionally, the technical specifications of the motors, caudal fin, elastic plate and springs should be provided.
2.The skin of the rear part of the body are removed, which will destroy the streamlined shape of the whole body, and lead to increase the drag dramatically.
3.It is best to provide some typical experimental videos about this robotic fish.
4.Only by comparing with the swimming performance of the real tuna can the advantages of the robot fish be understood.
5.How to determine the range of oscillating frequencies? The range of from 1 Hz to 4 Hz is also concerned about.
Author Response
We thank the reviewer for the comments and suggestions that helped us improve the manuscript. For convenience, all new parts of the text and rewritten paragraphs have been marked blue.
1. Since the dimensions of the robotic fish are not presented in this paper, it is very difficult to evaluate the swimming performance. Additionally, the technical specifications of the motors, caudal fin, elastic plate and springs should be provided.
Corrected. The specifications were added in the text of the manuscript and Fig. 1.
2. The skin of the rear part of the body are removed, which will destroy the streamlined shape of the whole body, and lead to increase the drag dramatically.
The tail section of the robot has been simplified since the task of the study was to test the new propulsion system in various modes of tail beat amplitude and frequency. We agree with the reviewer that this may lead to increase the drag. On the other hand this solution can provide a gain in energy consumption by reducing the mass and rigidity of the body part that is bent by the servo.
3. It is best to provide some typical experimental videos about this robotic fish.
A typical video of the floating robot is provided in the supplementary materials. In addition we've included snapshots of the movement of a real tuna vs. robot.
In case the video doesn't load on submission:
https://drive.google.com/file/d/1anDP3uuzOnN39BmpAuxaZWQMahJGUKXv/view
4. Only by comparing with the swimming performance of the real tuna can the advantages of the robot fish be understood.
We added remarks on this comparison in the conclusion. In this study we do not claim to achieve performance of real tuna which may be a very complicated and long task. Our study was focused on the dependence of performance on the propulsion parameters as just a step in modeling thuniform swimming.
5.How to determine the range of oscillating frequencies? The range of from 1 Hz to 4 Hz is also concerned about.
We added a new part in the section 3 concerning ranges of the kinematic parameters. By mistake, in the old version of the manuscript, we indicated double values of the tail beat frequency. We actually tested the 3.4 Hz, 2.5 Hz and 1.9 Hz modes. Corresponding corrections were made to the manuscript.
Round 2
Reviewer 1 Report
1. The innovation and contribution of the paper is unclear in the current version. Please clarify the scientific contribution made in this work, especially as opposed to the state-of-the-art of the robotic fish. Meanwhile, please provide solid supporting material.
2. There is no comparative experiment in the paper.
3. Please check and polish the paper thoroughly. For example, the first sentence of the abstract, “The paper describes a bioinspired propulsion system for an underwater robotic”, is problematic.
Author Response
We thank the reviewer for careful reading of the manuscript and questions that helped us improve the manuscript. Below, please find a point-by-point our responses on the questions and critical comments.
- The innovation and contribution of the paper is unclear in the current version. Please clarify the scientific contribution made in this work, especially as opposed to the state-of-the-art of the robotic fish. Meanwhile, please provide solid supporting material.
As proposed by the reviewer, we have expanded the description of our contributions and claims by revising the abstract, introduction and conclusion (for convenience, new inserts are marked with a yellow marker).
Specifically, we have found several new references related to our study. In the revised text we added brief descriptions of these studies comparing with our work. We added citations to new reviews [1-3] and took a closer look at robotic fish with muscle-mimicking drives [17-24]. We emphasized that our work was target to develop a simplified, low-cost, but bio-inspired propulsion system and also analyzed basic functionality of this robot. One of the interesting facts is that for fixed frequencies the increase of the oscillation amplitude led to the increase in the swimming speed only up to a certain threshold. It would be interesting to see in future if this threshold appears in robots of similar constructions developed by other groups. Indeed, each robot is unique and it would be great to find common critical facts in their functionality.
- There is no comparative experiment in the paper.
In our experiments, we compared the speed and cost of transport for different parameters of the propulsion system (frequency and amplitude). In the revised version with expanded reference list we also compared the results of our robot (speed and COT) with the results of other works.
In particular, the maximum speed of our robotic fish was about 0.4 BL/s, which is below the range of 0.58-2.15 BL/s shown by robots with wire-driven mover without tail simplification [18–20,22,23]. However, the presented robot demonstrated a speed higher than in other works (0.22 BL/s [26], 0.254 BL/s [27], 0.308 BL/s [28]), where a simplified version of the tail section was also used. Obviously, the advantages of a simplified design (the reduction of mass and rigidity of the body part bent by the servo drive) still cannot compensate for the increased drag caused by elevated turbulent flows. The loss in speed in this case can be considered a price for the simplicity and low cost of the robot.
We also found there is an interval of energetically preferable traveling speeds up to a threshold speed. Movement with higher speed was also possible but appeared to be more power-consuming. This conclusion qualitatively agreed with the data of COT studies from living tunas [4,29,30], although quantitatively the values obtained for the robot were still higher. Thus, the swimming efficiency of live tuna lies in the interval 1-7 J kg-1 m-1 depending on swimming speed. The values obtained for the presented robot were 30-70 J kg-1 m-1 for different values of frequency and swimming speed. The values obtained were also higher compared to the most outstanding results in this area: 4.06-11.80 J kg−1 m−1 [16] and 4.5-15.0 J kg−1 m−1 [3].
- Please check and polish the paper thoroughly. For example, the first sentence of the abstract, “The paper describes a bioinspired propulsion system for an underwater robotic”, is problematic.
We have checked and corrected the text. English was also corrected. For convenience, we have marked such edits with a green marker.
Reviewer 3 Report
No additional comments should be added.
Author Response
We thank the reviewer. We have checked and corrected the text. English was corrected. For convenience, we have marked such edits with a green marker.